# Oxidative Stress Causes Vacuolar Fragmentation in the Human Fungal Pathogen *Cryptococcus neoformans*

**DOI:** 10.3390/jof7070523

**Published:** 2021-06-29

**Authors:** Donghyeun Kim, Moonyong Song, Eunsoo Do, Yoojeong Choi, James W. Kronstad, Won Hee Jung

**Affiliations:** 1Department of Systems Biotechnology, Chung-Ang University, Anseong 17546, Korea; dfk94@cau.ac.kr (D.K.); smy5241@cau.ac.kr (M.S.); uo_oz@daum.net (Y.C.); 2Department of Microbiology, University of Georgia, Athens, GA 30602, USA; Eunsoo.Do@uga.edu; 3Michael Smith Laboratories, Department of Microbiology & Immunology, University of British Columbia, Vancouver, BC V6T 1Z4, Canada; kronstad@msl.ubc.ca

**Keywords:** *Cryptococcus neoformans*, fragmentation, oxidative stress, superoxide dismutase, vacuole

## Abstract

Vacuoles are dynamic cellular organelles, and their morphology is altered by various stimuli or stresses. Vacuoles play an important role in the physiology and virulence of many fungal pathogens. For example, a *Cryptococcus neoformans* mutant deficient in vacuolar functions showed significantly reduced expression of virulence factors such as capsule and melanin synthesis and was avirulent in a mouse model of cryptococcosis. In the current study, we found significantly increased vacuolar fragmentation in the *C. neoformans* mutants lacking *SOD1* or *SOD2*, which respectively encode Zn, Cu-superoxide dismutase and Mn-superoxide dismutase. The *sod2* mutant showed a greater level of vacuole fragmentation than the *sod1* mutant. We also observed that the vacuoles were highly fragmented when wild-type cells were grown in a medium containing high concentrations of iron, copper, or zinc. Moreover, elevated temperature and treatment with the antifungal drug fluconazole caused increased vacuolar fragmentation. These conditions also commonly cause an increase in the levels of intracellular reactive oxygen species in the fungus, suggesting that vacuoles are fragmented in response to oxidative stress. Furthermore, we observed that Sod2 is not only localized in mitochondria but also in the cytoplasm within phagocytosed *C. neoformans* cells, possibly due to copper or iron limitation.

## 1. Introduction

The vacuole is an important cellular organelle in fungi, and it performs diverse functions including the storage of ions, amino acids, phosphate, and carbohydrates. The vacuole morphology also dynamically adapts to intra- and extracellular environmental conditions through fusion (forming a single enlarged vacuole) and fission, which is also called fragmentation. For example, vacuole fusion is induced by nutrient limitation or hypotonic treatment, whereas vacuole fragmentation is triggered by hyperosmotic shock, endoplasmic reticulum stress, or lactic acid stress in the model fungus *Saccharomyces cerevisiae* [1,2]. The underlying mechanisms governing vacuole morphological changes are not fully understood. However, studies using *S. cerevisiae* suggest that increased vacuole volume by fusion enhances degradation of cellular materials through autophagy and multivesicular body pathways [1]. Moreover, genetically increased vacuole fusion extends the lifespan of *S. cerevisiae* [3]. Vacuole fragmentation is a complex process, and several proteins have been shown to be involved. Examples include Fab1, a lipid kinase that produces phosphatidylinositol 3,5-bisphosphate [PI(3,5)P_2_] from phosphatidylinositol 3-phosphate in *S. cerevisiae*. The *S. cerevisiae* mutant lacking the *FAB1* gene exhibits an enlarged vacuole, indicating the involvement of the protein in vacuole fragmentation [4]. In addition to Fab1, the target of rapamycin complex 1 (*TORC1*) is required for vacuole fragmentation in *S. cerevisiae* and cells treated with rapamycin do not display vacuole fragmentation [5]. The involvement of superoxide dismutase (SOD) in vacuole fragmentation has also been demonstrated [6]. Specifically, loss of *S. cerevisiae* Zn, Cu -superoxide dismutase Sod1, but not Mn-superoxide dismutase Sod2, led to significantly increased vacuole fragmentation [6]. SOD converts superoxide radicals into hydrogen peroxide and molecular oxygen. Therefore, increased vacuole fragmentation in the *sod1* mutant implied an influence of superoxide radicals on vacuole morphology and function in *S. cerevisiae*.

*Cryptococcus neoformans* is an opportunistic fungal pathogen that causes cryptococcal meningitis in immunocompromised hosts. *C. neoformans* possesses two genes, *SOD1* and *SOD2*, which are similar to the genes in *S. cerevisiae* and encode Zn, Cu-superoxide dismutase and Mn-superoxide dismutase, respectively. A number of biochemical and genetic studies have investigated the role of these enzymes in the physiology and virulence of *C. neoformans* [7,8,9,10,11]. For example, mutants lacking either *SOD1* or *SOD2* displayed increased susceptibility to oxidative stress and reduced growth at higher temperatures. Moreover, there was reduced survival of mice infected with the *sod1* mutant, whereas disease was completely abolished with the *sod2* mutant in a murine model of cryptococcosis, suggesting the importance of these enzymes in the pathogenesis of *C. neoformans*, although their contributions may be different [8,9].

Furthermore, a recent study revealed that the expression of *SOD1* and *SOD2* is directly regulated by Cuf1, the transcription factor that controls expression of the genes involved in copper homeostasis in *C. neoformans*. Indeed, the Cuf1 binding site (copper responsive elements) was identified within the promoter regions of both *SOD1* and *SOD2*. The same study also identified two isoforms of Sod2, one is the canonical mitochondrial protein, and the other is a novel cytosolic isoform specifically expressed under copper-limiting conditions, thus revealing Cuf1 and copper-dependent regulation of Sod1 and Sod2 in *C. neoformans* [12]. In addition to the association of SODs with copper homeostasis, we recently showed the influence of SODs on iron homeostasis, and on the cytosolic heme pool in particular, in *C. neoformans*. By utilizing a cytosolic heme sensor, we demonstrated that deletion of *SOD2* caused a significant reduction in intracellular heme levels due to increased ROS in the *sod2* mutant, whereas the *sod1* mutant displayed similar intracellular heme levels compared with the wild-type strain. In the same study, we also found that inhibition of vacuole function reduced intracellular heme levels, suggesting that the vacuole is an important organelle for maintaining heme homeostasis [13]. 

In this study, we addressed the role of SODs in *C. neoformans* in more detail, specifically focusing on iron homeostasis and vacuole morphology. The localization of both SODs was first confirmed by fusion with green fluorescent protein (GFP), and the influence of the enzymes on the major iron regulatory proteins, Cir1 and HapX, was investigated to understand the connection between the SOD proteins and iron homeostasis. We also studied the effect of deletion of *SOD1* and *SOD2* on vacuole morphology and the response to oxidative stress to determine the role of SODs in the physiology of *C. neoformans*. Finally, the mechanism by which Fab1 and TORC1 regulate vacuole morphology in *C. neoformans* was investigated.

## 2. Materials and Methods

### 2.1. Strains and Growth Media

*Cryptococcus neoformans* var. *grubii* H99 was used as the wild-type strain. The *sod1* and *sod2* mutants used in the current study were constructed as previously described [13]. For fungal cultures, yeast extract-bacto peptone medium supplemented with 2% glucose (YPD) was used. To prepare media with low concentrations of iron, copper, or zinc, 100 μM of bathophenanthrolinedisulfonic acid (BPS), bathocuproinedisulfonic acid disodium salt, or *N*,*N*,*N′*,*N′*-tetrakis(2-pyridinylmethyl)-1,2-ethanediamine (Sigma, Munich, BY, Germany) were added to YPD, respectively. Media with high concentrations of iron, copper, or zinc were prepared by supplementing low-iron, low-copper, or low-zinc media with 250 μM FeCl_3_, CuSO_4_, or ZnCl_2_, respectively.

### 2.2. Construction of Strains Expressing the Sod1-GFP or the Sod2-GFP Fusion Protein 

To construct the genes to express the Sod1-GFP or Sod2-GFP fusion proteins, the *SOD1* and *SOD2* genes were amplified by PCR using the wild-type genomic DNA and primers Sod1-GFP_F/Sod1-GFP_R, and Sod2-GFP_F/Sod2-GFP_R, respectively (Appendix A). The PCR fragments for *SOD1* and *SOD2* were digested with HindIII or BamHI, respectively, and cloned into pWH091 as previously described [14]. The resulting plasmids were named pWH151 and pWH152 and were digested with XmaI and BspHI, respectively. Digested pWH151 and pWH152 plasmids were transformed to the *sod1* or *sod2* mutants, respectively, and positive transformants were selected and confirmed by PCR and Western blot analysis.

### 2.3. Construction of the fab1 Knock-Out Mutant

To construct the *fab1* knock-out mutant, the genetic locus (CNAG_01209) containing the 8142-base pair open reading frame of the gene was replaced by homologous recombination with a gene-specific deletion cassette, which was amplified using the genomic DNA of the wild-type strain and pCH233, which contains the nourseothricin acetyltransferase gene as a template. The amplified cassette was transformed into the wild-type strain by biolistic transformation [15]. Positive transformants were confirmed using PCR. Two independent mutants were analyzed and found to have the same phenotypes.

### 2.4. Confocal Fluorescence Microscopy

To investigate the localization of Sod1 and Sod2, the strains expressing the Sod1-GFP or Sod2-GFP fusion proteins were grown in YPD medium at 30 °C overnight, diluted to an OD_600_ of 0.1 in fresh YPD medium, and incubated for 12 h. The cells were centrifuged at 5000 rpm for 5 min. The supernatant was discarded, and the cells were washed twice with phosphate-buffered saline (PBS) and suspended in 50 μL of PBS or PBS containing MitoTracker (Invitrogen, Waltham, MA, USA). Cells were visualized using a confocal fluorescence microscope LSM800Airy (Carl Zeiss, Oberkochen, BW, Germany) at a magnification of 1000×; images were obtained using the ZEN 3.2 blue edition software (Carl Zeiss, Oberkochen, BW, Germany).

To observe the vacuole, cells were grown in YPD medium overnight and diluted to an OD_600_ of 0.1 in fresh YPD medium or the same medium containing menadione, FeCl_3_, CuSO_4_, ZnCl_2_, or rapamycin as indicated in the figures. Cells were cultured to an OD_600_ of 1.0 and harvested by centrifugation at 5000 rpm for 5 min. The supernatant was discarded, and the cell pellet was suspended in 50 μL of YPD containing 0.5 μL of FM4-64 (a final concentration of 16 μM) (Thermo Fisher Scientific, Waltham, MA, USA). The cell suspension was incubated in the dark at 30°C for 30 min, centrifuged at 5000 rpm for 3 min, washed twice with PBS, and suspended in 50 μL of PBS. The morphology and number of vacuoles in the cells were determined by confocal fluorescence microscopy as described above. 

To observe localization of the Sod2-GFP fusion protein in phagolysosomes, murine macrophage-like cells (J774A.1) were maintained in Dulbecco’s modified Eagle medium (DMEM; GenDEPOT, Katy, TX, USA) with 10% fetal bovine serum (Atlas, Fort Collins, CO, USA) and 100 μg/mL penicillin-streptomycin (Gibco, Grand Island, NY, USA) at 37 °C in 5% CO_2_. Macrophages (5 × 10^5^ cells) were seeded to a 24-well plate containing coverslips and incubated at 37 °C in 5% CO_2_ overnight. *C. neoformans* cells were incubated in YPD at 30 °C overnight and opsonized with 9.34 μg/mL monoclonal 18B7 antibody at 37 °C for one hour. Macrophages were activated with 100 nM of phorbol 12-myritate 13-acetate (Sigma, St. Louis, MO, USA) at 37 °C in 5% CO_2_ for one hour. Opsonized *C. neoformans* cells were coincubated with activated macrophages at 37 °C in 5% CO_2_. After one hour, each well of the 24-well plate was washed three times with phosphate buffered saline (PBS) to remove non-phagocytized fungal cells, and the plate was incubated in fresh DMEM at 37 °C for 18 h. For fluorescence microscopic observation, a coverslip was removed from each well of the plate, washed five times with PBS, mounted on a glass slide using Prolong™ Gold Antifade Reagent (Invitrogen, Waltham, MA, USA), and localization of the Sod2-GFP protein in phagocytized cells was observed. In parallel, macrophages were also lysed by addition of sterilized deionized water, phagocytized *C. neoformans* cells were isolated, and localization of the Sod2-GFP was observed.

### 2.5. Flow Cytometric Analysis

Cellular ROS levels were determined using MitoSOX-based assays and flow cytometric analysis [16]. Cells were grown in YPD medium at 30 °C overnight, diluted to an OD_600_ of 0.1 in fresh YPD medium and incubated at 30 °C for 12 h. The cells were collected, washed twice with fresh YPD medium, resuspended in YPD medium containing 5 μM MitoSOX (Thermo Fisher Scientific, Waltham, MA, USA), and incubated in the dark at 30 °C for 30 min. The cells were washed twice with PBS and suspended in 5 mL PBS. Flow cytometric analysis was performed using FACS AriaII (BD Bioscience, San Jose, CA, USA); data were analyzed using Flowing software (Turku Bioscience, Turku, Finland).

### 2.6. Western Blot Analysis

Cells were grown in YPD medium at 30 °C overnight and diluted to an OD_600_ of 0.1, in YPD medium containing 150 μM BPS to deplete iron. Cultured cells were harvested, diluted to an OD_600_ of 0.1 in YPD medium containing 150 μM BPS with or without 100 μM FeCl_3_, and incubated at 30 °C for 12 h. Cells were harvested by centrifugation and resuspended in a protein extraction buffer comprised of 50 mM HEPES KOH pH 7.5, 140 mM NaCl, 1 mM EDTA, 1% Triton X-100, 0.1% Na-deoxycolate, 1 mM PMSF, and a protease inhibitor cocktail (Sigma, Munich, BY, Germany). Cell lysates were prepared by bead-beating, and the protein concentration was determined by the Bradford assay [17]. Total proteins were separated on a sodium dodecyl sulfate–polyacrylamide gel and transferred to a nitrocellulose membrane (Sigma, Munich, BY, Germany). Protein detection was performed using an anti-GFP rabbit polyclonal antibody (Proteintech, Rosemont, IL, USA) or an anti-FLAG (Abcam, Cambridge, UK) rabbit polyclonal antibody. A mouse anti-rabbit IgG horseradish peroxidase conjugate (Santa Cruz, Dallas, TX, USA) was used as a secondary antibody, followed by visualization by chemiluminescence.

### 2.7. Superoxide Dismutase Activity Assay

Superoxide dismutase activity assay was performed as described previously [7]. Briefly, cells were incubated in YPD containing 150 µM BPS with or without 250 µM FeCl3 at a concentration of 1 × 10^7^ cells/mL for 12 h. The cells were harvested and lysed using native lysis buffer (50 mM Tris-Cl pH 7.5, 0.1 mM EDTA, 1 mM PMSF). The 20 µg of total protein was separated by native gel electrophoresis (132 mM Tris-base, 132 mM borate, 3.6 mM sodium citrate, 0.1% ammonium persulfate) and stained with nitro blue tetrazolium at room temperature for 30 min in dark. The gel was illuminated for 5 min and SOD activities in the wild-type strain, the *sod1* mutant, and the *sod2* mutant were visualized (Appendix A).

## 3. Results

### 3.1. Localization of Sod2 Is Influenced by Iron and Copper, and Iron Levels Impact the Expression and Activity of Sod2

We initially constructed strains expressing the Sod1-GFP or the Sod2-GFP fusion protein (Section 2) to investigate localization of the proteins. The functionality of the fusion proteins was confirmed by growing the cells under high concentrations of NaCl and in the presence of a superoxide anion generating agent menadione, the conditions of which reduced the growth of the *sod1* and *sod2* mutants [7,18]. As shown in Figure 1A, the strains expressing the Sod1-GFP or the Sod2-GFP fusion protein grew as well as the wild-type strain thus indicating the fusion proteins are fully functional. Examination of the intracellular localization of the fusion proteins by fluorescence microscopy revealed that Sod1-GFP and Sod2-GFP were localized in the cytoplasm and mitochondria, respectively, when the cells were grown in rich-medium (Figure 1B).

Recently, Smith et al. demonstrated a cytosolic isoform of the Sod2 protein upon copper limitation [12]. This led us to investigate localization of Sod2-GFP in the host environment of the phagolysosome of a murine macrophage-like cell line, where the metal concentrations may be significantly limited. Cells of the *C. neoformans* strain expressing the Sod2-GFP protein were phagocytized and localization of the protein in the fungal cells within phagolysosomes was analyzed. Our results indicated that Sod2-GFP localized not only in mitochondria but also in the cytosol in internalized *C. neoformans* cells (Figure 1C). We noted that not all phagocytized cells displayed cytosolic localization of the Sod2-GFP protein; the proportion of cells displaying this phenotype was 67%. To confirm that cytosolic localization of the Sod2-GFP fusion protein was influenced by copper limitation, we investigated location of the protein in cells grown in medium containing a copper chelator. Indeed, copper limitation resulted in partial cytosolic localization of the Sod2-GFP fusion protein. Furthermore, partial cytosolic localization of the protein was also observed upon iron limitation (Figure 1D). 

Superoxide dismutase is required for maintaining intact iron-sulfur (Fe-S) clusters and iron homeostasis, and its expression is regulated by intra- and extracellular iron levels. For example, cytosolic Zn, Cu-superoxide dismutase SodA expression is upregulated upon iron depletion in both *Aspergillus nidulans* and *A. fumigatus* [19]. Additionally, increased expression of the cell wall-associated Cu-superoxide dismutase, Sod4, was also observed in *C. albicans* upon iron depletion [20]. In this context, we investigated whether the expression of the SOD proteins in *C. neoformans* was influenced by iron concentrations in the medium. Our results showed that the levels of Sod1 were generally higher than those of Sod2 and were independent of iron concentration in the medium. However, the level of Sod2 protein was significantly increased in the medium containing a high concentration of iron (Figure 2A). These results suggest an association between Sod2 as a mitochondrial and cytosolic Mn-SOD, and iron homeostasis in *C. neoformans*. However, our results were different from those observed with *Aspergillus* and *Candida*, suggesting that Sod2 is the major SOD and is significantly influenced by iron levels in *C. neoformans*. Moreover, our conclusion was further supported by the observation of significantly increased Sod2 activity in wild-type or *sod1* mutant cells grown in the presence of a high concentration of iron (Figure 2B). 

### 3.2. Deletion of SOD Influences Expression of Iron Regulators as Well as Vacuole Morphology

We next characterized the phenotypes of mutants lacking *SOD1* or *SOD2* (constructed in our recent study) [13]. Growth analysis showed that the mutant lacking *SOD1* grew like the wild-type strain in YPD medium (data not shown), but the mutant lacking *SOD2* displayed a reduced doubling time (Figure 2C) (WT: 84.6 min; *sod2**Δ*: 93.6 min). We reasoned that the slower growth of the *sod2* mutant may be caused mainly by failure to handle oxidative stress, and therefore we measured ROS accumulation in the cells. We found that compared with the wild-type strain, the *sod2* mutant displayed significantly increased intracellular ROS levels. The *sod1* mutant also showed increased ROS levels, but to a lesser extent than the *sod2* mutant (Figure 2D).

Increased expression and activity of Sod2 in the wild-type cells at an elevated iron concentration and elevated ROS levels in the mutant implied that the *sod2* mutant is deficient in iron homeostasis. Therefore, we investigated the expression of Cir1 and HapX, the major iron transcription factors in *C. neoformans*, in the *sod2* mutant. Previously, it was shown that iron increased the stability and abundance of Cir1 in wild-type cells [21]. In the current study, we confirmed increased Cir1 protein levels in wild-type cells growing under high-iron conditions. However, levels of Cir1 were significantly reduced in the *sod2* mutant, especially in the high iron condition, indicating that Sod2 influences the abundance of Cir1 (Figure 2E). The influence on HapX protein levels was more dramatic because deletion of *SOD2* reduced the abundance of the protein in both high- and low-iron conditions (Figure 2F). These results suggest that Sod2 is critical for maintaining proper iron regulation mediated by the major iron regulators, Cir1 and HapX, although the underlying mechanism(s) require further investigation.

Vacuole fragmentation was one of the key phenotypes observed in *S. cerevisiae SOD* mutants, particularly for the *sod1* mutant [6]. Accordingly, we investigated the morphology of the vacuoles in the *C. neoformans* mutants lacking *SOD1* or *SOD2* by confocal fluorescence microscopy. While the majority of the wild-type cells showed a single vacuole, an average of two vacuoles per cell was observed in the *sod1* mutant. In contrast, the number of vacuoles was dramatically increased in the *sod2* mutant, with an average of four vacuoles per cell, resembling vacuole fragmentation in *S. cerevisiae* (Figure 3). These results suggest that activity of both SODs is instrumental in preventing vacuolar fragmentation, but Sod2 plays a more important role in maintaining the proper morphology of the vacuole in *C. neoformans*. 

### 3.3. Oxidative Stress Induces Vacuole Fragmentation

One of the key roles of SODs in cells is to neutralize ROS, and mutants lacking SOD therefore normally show increased susceptibility to various oxidative stresses. Sod2 has previously been shown to mediate protection against oxidative stress in *C. neoformans* [12]. Here, we observed that deletion of *SOD1* or *SOD2* (to a greater extent) increased intracellular ROS levels and caused increased susceptibility to oxidative stress. Therefore, we reasoned that vacuole fragmentation would also be induced by increased oxidative stress. To test this hypothesis, we induced oxidative stress in wild-type cells by growth in medium containing menadione and subsequently analyzed the vacuole morphology of the cells. This approach revealed significantly increased vacuole fragmentation in cells treated with menadione confirming that oxidative stress induced vacuole fragmentation in *C. neoformans* (Figure 4). 

As mentioned above, we observed that an elevated concentration of iron increased the expression of Sod2, which suggested that this metal causes an oxidative stress response and increases Sod2 protein expression. To establish whether excess iron causes vacuole fragmentation, wild-type cells were grown in a medium containing high concentrations of iron, and their vacuoles were observed. Copper and zinc were also tested. The results showed that high concentrations of each of the metals increased vacuole fragmentation (Figure 5A). Moreover, our results indicated that a main contributor to metal-mediated vacuole fragmentation was the increase in ROS levels in the cells treated with high concentrations of the metals (Figure 5B).

Growing *C. neoformans* cells at 37 °C—the host body temperature—is known to induce oxidative stress [22]. Additionally, the *sod2* mutant also displays a severe growth defect at 37 °C, presumably due to increased susceptibility of the cells to ROS generated at this temperature [9]. *C. neoformans* cells also suffer from oxidative stress when they are treated with an azole antifungal drug, such as fluconazole, which not only inhibits ergosterol biosynthesis but also increases the production of ROS in the fungus [23,24]. Therefore, we hypothesized that growing wild-type cells at 37 °C or treating them with fluconazole would cause oxidative stress and, in turn, increase vacuole fragmentation. To test this idea, we cultured the cells at 37 °C or in the presence of fluconazole and found that both conditions increased vacuole fragmentation (Figure 5C). Furthermore, significantly increased intracellular ROS levels were observed in the cells grown under these conditions, a finding consistent with oxidative stress causing vacuole fragmentation (Figure 5D).

### 3.4. Vacuole Fragmentation Is Also Influenced by Fab1 and TORC1

As mentioned previously, Fab1 is involved in vacuole fragmentation in *S. cerevisiae* [4]. Therefore, we investigated the influence of the Fab1 homolog on Sod2-mediated vacuole fragmentation in *C. neoformans*. Specifically, we generated a *fab1* mutant and investigated vacuole morphology (Figure 6). The *fab1* mutant possessed a single but enlarged vacuole per cell upon growth in YPD medium, and its vacuole morphology was unaffected by the addition of metals, in contrast to the observations for wild-type cells. These results suggest that Fab1 is an upstream component that influences vacuole fragmentation in *C. neoformans* (Figure 6). In addition to Fab1, TORC1 is also required for vacuole fragmentation in *S. cerevisiae* [5,25]. Therefore, we tested whether TORC1 also influences vacuole fragmentation in *C. neoformans* by growing wild-type and *sod2* mutant cells with rapamycin. Wild-type *S. cerevisiae* cells were included as controls. The results showed that, similar to *S. cerevisiae* cells, vacuole fragmentation was significantly inhibited by rapamycin in both the wild-type and the *sod2* mutant cells in either YPD or high-iron medium (Figure 7). This suggests that, similar to *S. cerevisiae*, the TOR signaling pathway regulates vacuole fragmentation in *C. neoformans*, likely with Fab1. Although we observed the involvement of Fab1 and TORC1 in vacuole fragmentation in *C. neoformans*, the mechanism by which SODs affect the function of these proteins still needs to be explored.

## 4. Discussion

Trace metals such as iron, zinc, and copper are essential nutrients for every organism. However, excess amounts of these metals generate ROS and have a detrimental effect on the cell. In the current study, we found that high concentrations of iron as well as zinc and copper induced vacuole fragmentation in wild-type *C. neoformans* cells likely because these metals increased intracellular ROS accumulation. Iron overload is known to cause the production of reactive free radicals via the Fenton reaction [26,27,28]. Therefore, the high concentration of iron in the medium may have increased the intracellular labile iron pool, which in turn generated ROS. Excess zinc can also cause an increase in ROS accumulation, and disrupt iron metabolism, such as Fe-S cluster synthesis, in *S. cerevisiae* [29,30]. Excess copper also generates ROS and causes oxidative stress, DNA damage, and membrane disruption in *S. cerevisiae* [31,32]. Furthermore, similar to the excess amount of zinc, Fe-S cluster synthesis is inhibited by copper toxicity [33]. Overall, cells possess multiple mechanisms to maintain intracellular metal homeostasis that can be disrupted by elevated ROS. 

In *C. neoformans*, iron homeostasis has garnered much attention because of its involvement in the expression of virulence factors such as the polysaccharide capsule [34,35]. Cir1 and HapX are two well-characterized iron regulatory transcription factors in *C. neoformans*. The protein abundance of Cir1 and HapX are both influenced by iron levels. Under high-iron conditions, Cir1 is more abundant, and the protein acts as a repressor for genes, including those involved in iron uptake and metabolism. In contrast, HapX mainly acts as a repressor under low-iron conditions to inhibit the expression of genes involved in iron-consuming functions [36]. In the current study, we found that the protein levels of both Cir1 and HapX were significantly reduced in the *sod2* mutant. Of these changes, reduction of Cir1 might trigger a more detrimental effect on the mutant cells because of the de-repression of genes involved in iron uptake, causing increased accumulation of intracellular iron. In mammalian cells, overexpression of mitochondrial Mn-SOD significantly reduced non-heme iron levels, which is in agreement with our data [37,38]. We hypothesize that increased superoxide due to lack of Sod2 may also cause a reduction in the stability and activity of Cir1 and HapX since the proteins are predicted to contain Fe-S clusters and iron influences their stability [21,36]. Indeed, previous studies have reported that superoxide causes the release of iron from the Fe-S cluster and in turn inactivates the Fe-S cluster-containing protein [39,40].

We found that oxidative stress induces vacuole fragmentation in *C. neoformans*, and deletion of *SOD2* exaggerated fragmentation of the organelle. A similar phenomenon was observed in *S. cerevisiae*, although vacuole fragmentation was mainly observed under hyperosmotic conditions in relation to the transient increase in intracellular levels of PI(3,5)P_2_, which was synthesized by Fab1 and presented at the vacuole membrane [2,41]. Loss of water under hyperosmotic conditions was hypothesized to reduce the volume of vacuoles and trigger fragmentation of the organelle, and consequently, increase levels of PI(3,5)P_2_ [2]. In addition to hyperosmotic conditions, oxidative stress also induced vacuole fragmentation in *S. cerevisiae*. However, unlike *C. neoformans*, the mutant lacking cytosolic *SOD1*, but not mitochondrial *SOD2*, showed increased vacuole fragmentation in *S. cerevisiae* [6]. Moreover, vacuole fragmentation in the *S. cerevisiae sod1* mutant was reduced upon induction of iron deficiency and increased by the addition of excess metal. The vacuole is the main iron storage organelle in *S. cerevisiae*, and at least two vacuolar iron transport systems have been identified in the fungus. One is mediated by the homologs of the high-affinity plasma membrane iron transport system Fth1-Fet5, whereas the other is mediated by the iron/Mn^2+^ transporter, Ccc1 [42,43]. Iron storage may be disrupted by the deletion of *SOD1* due to defects in correct vacuole morphology in *S. cerevisiae* [6]. We also speculate that, based on the previous studies and our current findings, vacuole fragmentation increases the surface-to-volume ratio and provides more docking sites for vacuolar iron transporters to increase iron uptake into vacuoles to protect cells from oxidative stress. A positive correlation between the surface-to-volume ratio and increased activity of vacuole transporters was also hypothesized by Michaillat et al. [5]. In general, the regulation of vacuole morphology for *C. neoformans* may be similar to that of *S.*
*cerevisiae*. However, Sod2, rather than Sod1, plays a more critical role in vacuole morphology in *C. neoformans*. Overall, we conclude that the function and morphology of vacuoles are tightly connected to metal homeostasis and oxidative stress responses in *C. neoformans*.

TORC1 is a protein complex that consists of Tor1 or Tor2, Tco89, Kog1, and Lst8, which are mainly localized at the membrane of the vacuoles in *S. cerevisiae*, and the involvement of TORC1 in vacuole morphology has been documented [44]. The TORC1 pathway has been identified in *C. neoformans*, and a study confirmed its main localization at the vacuole membrane [45]. Moreover, the influence of the TORC1 pathway on thermotolerance and the DNA damage response in *C. neoformans* has been suggested [45]. Here, we showed that rapamycin treatment reduced vacuole fragmentation in *C. neoformans*, suggesting a role for TORC1 in the vacuole morphology of the fungus. Moreover, our study demonstrated that TORC1 and the downstream target protein Fab1 may coordinately regulate vacuole morphology. Interestingly, TORC1 and Fab1 regulation is upstream of oxidative stress-mediated vacuole fragmentation in *C. neoformans* because the wild-type and the *sod2* mutant cells treated with rapamycin under the high concentrations of metals displayed a single large vacuole in our study. 

In summary, our data demonstrate that the vacuole morphology is significantly influenced by oxidative stress in *C. neoformans* and that SODs play an important role in mediating the response in the context of metal homeostasis. These aspects of the response to stress are important because an oxidative attack on *C. neoformans* cells is a key aspect of the innate immune defense mechanisms of host phagocytic cells. This idea is reinforced by our demonstration that a cytosolic localization of Sod2 occurred in phagocytosed *C. neoformans* cells, a phenotype that may be associated with metal limitation within a macrophage. 

## Figures and Tables

**Figure 1 jof-07-00523-f001:**
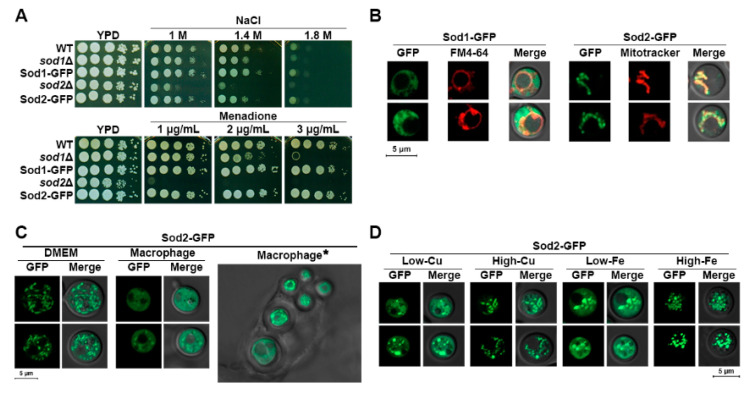
Localization of SODs. (**A**) Functionality of the Sod1-GFP and the Sod2-GFP fusion proteins was tested in spot assays of the strains on the medium containing NaCl or menadione. (**B**) Sod1 and Sod2 were each expressed as fusion proteins with GFP, and the respective cytosolic and mitochondrial localization of Sod1-GFP and Sod2-GFP was confirmed by fluorescent microscopy. FM4-64 and MitoTracker were used to visualize vacuoles and mitochondria, respectively. Images of two representative cells are presented for each strain. (**C**) Fungal cells grown in DMEM media showed mitochondrial localization of the Sod2-GFP protein (DMEM) while the cells isolated from the phagolysosome of the infected macrophages displayed predominantly cytosolic localization of the protein (Macrophage). The cryptococcal cells within a phagolysosome of a macrophage also showed cytosolic localization of the Sod2-GFP protein (Macrophage*). (**D**) Fluorescent microscopy of Sod2-GFP localization in cells grown in low-copper (DMEM containing 50 μM of bathocuproine disulfonate (BCS)) or high-copper (DMEM containing 50 μM BCS and 100 μM of CuSO_4_) medium, or low-iron (DMEM containing 50 μM of bathophenanthroline disulfonic acid (BPS)) or high-iron (DMEM containing 50 μM of BPS and 100 μM of FeCl_3_) medium.

**Figure 2 jof-07-00523-f002:**
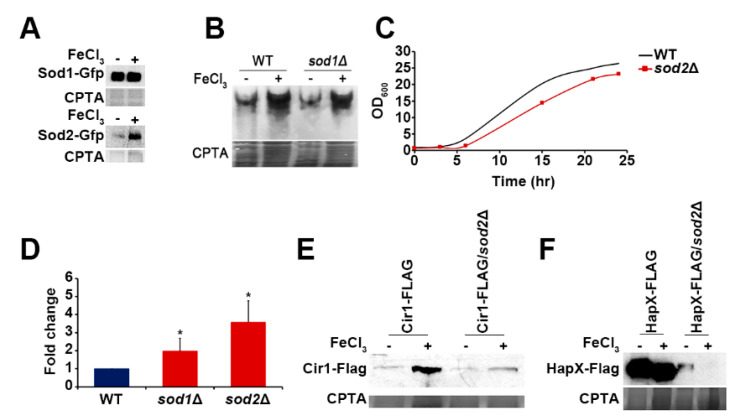
Protein levels and activity of Sod2 are influenced by iron, and deletion of *SOD2* caused reduced Cir1 and HapX levels. (**A**) Strains expressing Sod1-GFP or Sod2-GFP were grown in media with (+) or without (−) 100 μM FeCl_3_, and Western blot analysis was performed using total cell lysates. The membrane was stained with copper phthalocyanine-3,4′,4″,4‴-tetrasulfonic acid tetrasodium (CPTA) to confirm equal loading of each sample. (**B**) The Sod2 activities of the strains grown in media with (+) or without (−) 100 μM FeCl_3_ were measured by native gel electrophoresis and nitroblue tetrazolium staining. (**C**) The growth of the wild-type and the *sod2* mutant were monitored in YPD medium. (**D**) Cellular ROS levels in the strains were determined using the MitoSOX-based assays. ROS levels of the *sod1* and *sod2* mutants were compared with those of the wild-type strain. Averages from three independent experiments are presented with standard deviations (* *p* ≤ 0.05). (**E**) Strains expressing Cir1-FLAG in the wild-type or the *sod2* mutant were grown in media with (+) or without (−) 100 μM FeCl_3_; Western blot analysis was performed and the membrane was stained with CPTA. (**F**) Strains expressing HapX-FLAG in the wild-type or the *sod2* mutant were grown in media with (+) or without (−) 100 μM FeCl_3_; Western blot analysis was performed and the membrane was stained with CPTA.

**Figure 3 jof-07-00523-f003:**
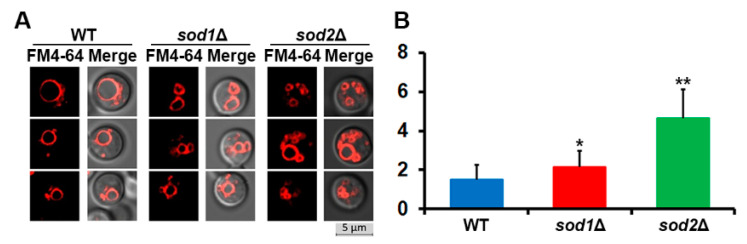
Deletion of *SOD1* or *SOD2* caused vacuole fragmentation. (**A**) Cells were grown in YPD medium, stained with FM4-64, and their vacuole morphology was analyzed by confocal fluorescence microscopy. (**B**) The number of vacuoles per cell of each strain (the vertical axis) was quantified. Averages from 50 independent cells of each strain are presented with standard deviations (* *p* ≤ 0.05; ** *p* ≤ 0.001).

**Figure 4 jof-07-00523-f004:**
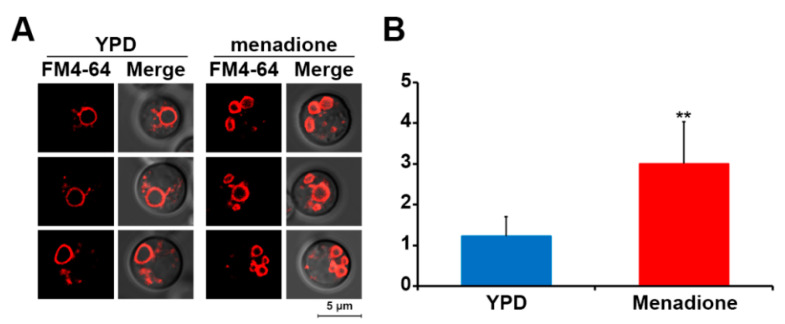
Oxidative stress increased vacuole fragmentation. (**A**) Wild-type cells were grown in YPD medium with or without menadione (0.1 μg/mL), stained with FM4-64, and their vacuole morphology was analyzed by confocal fluorescence microscopy. (**B**) The number of the vacuoles per cell of each strain (the vertical axis) was quantified. Averages from 50 independent cells of each strain are presented with standard deviations (** *p* ≤ 0.001).

**Figure 5 jof-07-00523-f005:**
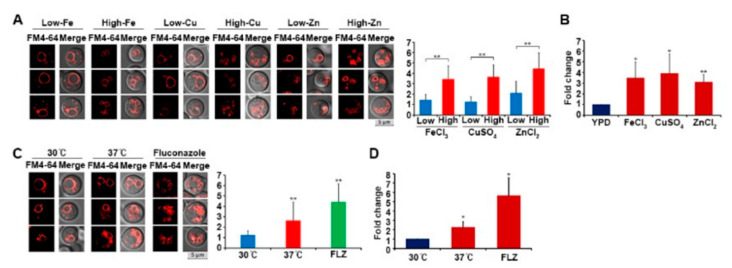
High concentration of iron, copper, or zinc, and high temperature or fluconazole treatment, induced vacuole fragmentation. (**A**) Wild-type cells were grown in media with low or high levels of metals (250 μM FeCl_3_, CuSO_4_, or ZnCl_2_). Cells were stained with FM4-64, and their vacuole morphology was analyzed by confocal fluorescence microscopy. Three independent cells from three independent experiments are presented. The bar graph presents the number of vacuoles per cell (the vertical axis) in each condition based on the averages from 50 independent cells of each strain with standard deviations (** *p* ≤ 0.001). (**B**) Cellular ROS levels were determined using MitoSOX-based assays. Averages from three independent experiments are presented with standard deviations (* *p* ≤ 0.05; ** *p* ≤ 0.001). (**C**) Wild-type cells were grown at 37 °C or in medium containing fluconazole (FLZ) as indicated. Cells were stained with FM4-64, and their vacuole morphology was analyzed using confocal fluorescence microscopy. Three independent cells from three independent experiments are presented. The bar graph presents the number of vacuoles per cell (the vertical axis) in each condition based on the averages from 50 independent cells of each strain with standard deviations (** *p* ≤ 0.001). (**D**) Cellular ROS levels in the strains grown at 37 °C or in medium containing fluconazole were determined using MitoSOX-based assays. The average of the values from three independent experiments with standard deviations are presented (* *p* ≤ 0.05).

**Figure 6 jof-07-00523-f006:**
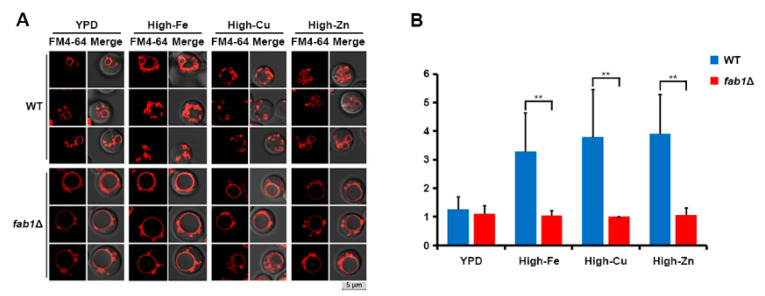
Fab1 influences vacuole fragmentation. (**A**) Wild-type and *fab1* mutant cells were grown in high-metal media containing 250 μM of FeCl_3_, CuSO_4_, or ZnCl_2_, stained with FM4-64, and their vacuole morphology was analyzed by confocal fluorescence microscopy. Three independent cells from three independent experiments are presented. (**B**) The number of vacuoles in a cell grown in each condition was counted (the vertical axis), and averages from 50 independent cells of each strain are presented with standard deviations (** *p* ≤ 0.001).

**Figure 7 jof-07-00523-f007:**
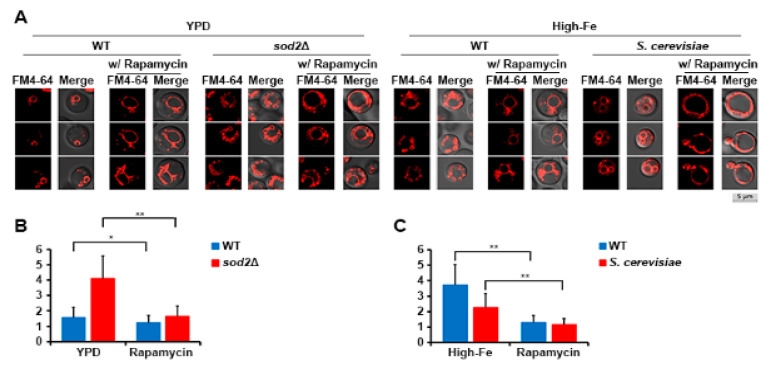
Vacuole fragmentation is inhibited by rapamycin. (**A**) Cells were grown in YPD or high-iron media with or without rapamycin, stained with FM4-64, and their vacuole morphology was analyzed using confocal fluorescence microscopy. Three independent cells from three independent experiments are presented. (**B**,**C**). The number of vacuoles in a cell grown in each condition was counted, and averages from 50 independent cells of each strain are presented with standard deviations (* *p* ≤ 0.05; ** *p* ≤ 0.001). *S. cerevisiae* BY4741 was included as a control. The vertical axis represents the number of vacuoles. Results present the average of the values from three independent experiments.

## Data Availability

Not applicable.

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
