# Peer review of "Oxidative Stress Causes Vacuolar Fragmentation in the Human Fungal Pathogen Cryptococcus neoformans"

_jof, 2021, doi:10.3390/jof7070523_

Round 1

Reviewer 1 Report

  • A brief summary - The authors investigated influence of superoxide dismutase (SOD) on vacuolar morphology and iron homeostasis as well as mechanism by which Fab1 and TORC1 regulate vacuole morphology in Cryptococcus neoformans. Significantly increased vacuolar fragmentation in the C. neoformans mutants lacking SOD1 or SOD2 was found as well as in induced oxidative stress. The authors also observed the involvement of Fab1 and TORC1 in vacuole fragmentation.
  • Broad comments - The strength of this study is possible future contribution to better understanding of immunologic response to C. neoformans especially of host phagocytic cells. There are no specific weaknesses in this manuscript that could be emphasized.
  • Specific comments - The content overall including all figures is very well presented and completely comprehensive. 

Author Response

We thank the reviewers for their comments on our manuscript. As listed below, we have addressed each of the comments from the reviewers and have improved the manuscript accordingly.

Reviewer 1:

Comments and Suggestions for Authors:

A brief summary - The authors investigated influence of superoxide dismutase (SOD) on vacuolar morphology and iron homeostasis as well as mechanism by which Fab1 and TORC1 regulate vacuole morphology in Cryptococcus neoformans. Significantly increased vacuolar fragmentation in the C. neoformans mutants lacking SOD1 or SOD2 was found as well as in induced oxidative stress. The authors also observed the involvement of Fab1 and TORC1 in vacuole fragmentation.

Broad comments - The strength of this study is possible future contribution to better understanding of immunologic response to C. neoformans especially of host phagocytic cells. There are no specific weaknesses in this manuscript that could be emphasized.

Specific comments - The content overall including all figures is very well presented and completely comprehensive.

>> Response: We thank the reviewer for the positive comments.

Reviewer 2 Report

The manuscript by Kim et al explores the connection between oxidative stress, superoxide dismutases and vacuolar fragmentation in Cryptococcus neoformans. The study shows that vacuolar fragmentation is triggered in response to the increase in intracellular ROS, and is alleviated by superoxide dismutases Sod1 and Sod2. Excess metals is a known trigger for ROS accumulation, and Sod2 was found to affect the abundance of two major iron regulatory proteins, Cir1 and HapX. TORC1 and lipid kinase Fab1 were also implicated, with inhibition of TORC1 leading to a decrease in a number of vacuoles.

Minor corrections:

  1. Mention of the previous work characterizing superoxide dismutases in Cryptococcus neoformans has to be included in the Introduction.

Characterization of Cu,Zn superoxide dismutase (SOD1) gene knock-out mutant of Cryptococcus neoformans var. gattii: role in biology and virulence. Narasipura SD, Ault JG, Behr MJ, Chaturvedi V, Chaturvedi S. Mol Microbiol. 2003 Mar;47(6):1681-94. doi: 10.1046/j.1365-2958.2003.03393.x. PMID: 12622821

Characterization of Cryptococcus neoformans variety gattii SOD2 reveals distinct roles of the two superoxide dismutases in fungal biology and virulence. Narasipura SD, Chaturvedi V, Chaturvedi S. Mol Microbiol. 2005 Mar;55(6):1782-800. doi: 10.1111/j.1365-2958.2005.04503.x. PMID: 15752200

  1. Line 220. Sod2 activity is shown to be increased only in WT, the data for sod1 mutant is missing.
  2. Line 250. The method for native gel electrophoresis and nitroblue tetrazolium staining is neither referenced nor described. It would also be beneficial to include the controls for this experiment, to show how Sod2 activity was differentiated from the Sod1 activity.
  3. Lines 258-259. “…elevated ROS levels in the mutant at an elevated iron concentration…” Corresponding method only mentions YPD as the medium, no added metals. This is an important detail potentially affecting the conclusions.
  4. Line 278. “…both SODs are involved in vacuole fragmentation”. Since vacuolar fragmentation occurs in SOD mutants, it would be correct to state “activity of both SODs is instrumental in preventing vacuolar fragmentation”.
  5. Figure 7. It is not immediately clear that WT refers to C. neoformans.
  6. Lines 438-441. Wouldn’t it be a reasonable conclusion that TORC1/Fab1 promote vacuolar fragmentation downstream of ROS signal since rapamycin blocks ROS-induced vacuolar fragmentation?

Author Response

We thank the reviewers for their comments on our manuscript. As listed below, we have addressed each of the comments from the reviewers and have improved the manuscript accordingly.

Reviewer 2:

Comments and Suggestions for Authors

The manuscript by Kim et al explores the connection between oxidative stress, superoxide dismutases and vacuolar fragmentation in Cryptococcus neoformans. The study shows that vacuolar fragmentation is triggered in response to the increase in intracellular ROS, and is alleviated by superoxide dismutases Sod1 and Sod2. Excess metals is a known trigger for ROS accumulation, and Sod2 was found to affect the abundance of two major iron regulatory proteins, Cir1 and HapX. TORC1 and lipid kinase Fab1 were also implicated, with inhibition of TORC1 leading to a decrease in a number of vacuoles.

Minor corrections:

Mention of the previous work characterizing superoxide dismutases in Cryptococcus neoformans has to be included in the Introduction.

Characterization of Cu,Zn superoxide dismutase (SOD1) gene knock-out mutant of Cryptococcus neoformans var. gattii: role in biology and virulence. Narasipura SD, Ault JG, Behr MJ, Chaturvedi V, Chaturvedi S. Mol Microbiol. 2003 Mar;47(6):1681-94. doi: 10.1046/j.1365-2958.2003.03393.x. PMID: 12622821

Characterization of Cryptococcus neoformans variety gattii SOD2 reveals distinct roles of the two superoxide dismutases in fungal biology and virulence. Narasipura SD, Chaturvedi V, Chaturvedi S. Mol Microbiol. 2005 Mar;55(6):1782-800. doi: 10.1111/j.1365-2958.2005.04503.x. PMID: 15752200

>>Response: References were added. Line 58.

Line 220. Sod2 activity is shown to be increased only in WT, the data for sod1 mutant is missing.

>> Response: The original image contains the results of the sod1 mutant. However, the sod1 mutant was excluded because we thought it was unnecessary. The Figure 2B was replaced with the original image that contains the sod1 mutant as recommend.  

Line 250. The method for native gel electrophoresis and nitroblue tetrazolium staining is neither referenced nor described. It would also be beneficial to include the controls for this experiment, to show how Sod2 activity was differentiated from the Sod1 activity.

>> Response: The assay method and reference were added, line 184-193. We think that the Sod1 signal (activity) was very weak on a nitroblue tetrazolium staining gel. As shown in the gel image below, the upper band was disappeared in the sod2 mutant indicating that the band represents the Sod2 activity. This image was included as a Supplementary data (Figure S1).

Lines 258-259. “…elevated ROS levels in the mutant at an elevated iron concentration…” Corresponding method only mentions YPD as the medium, no added metals. This is an important detail potentially affecting the conclusions.

>> Response: There was typo. The text changed to ‘Increased expression and activity of Sod2 in the wild-type cells at an elevated iron concentration, and elevated ROS levels in the mutant implied that the sod2 mutant is deficient in iron homeostasis’. Line 281-283.

Line 278. “…both SODs are involved in vacuole fragmentation”. Since vacuolar fragmentation occurs in SOD mutants, it would be correct to state “activity of both SODs is instrumental in preventing vacuolar fragmentation”.

>> Response: Changed accordingly. Line 301-302.

Figure 7. It is not immediately clear that WT refers to C. neoformans.

>> Response: Labels in the figure were changed (‘WT’ to ‘C. neoformans WT’)

Lines 438-441. Wouldn’t it be a reasonable conclusion that TORC1/Fab1 promote vacuolar fragmentation downstream of ROS signal since rapamycin blocks ROS-induced vacuolar fragmentation?

>> Response: Our data imply that TORC1 and Fab1 regulation is upstream of oxidative stress-mediated vacuole fragmentation in C. neoformans. However, at the moment, we do not have any direct experimental evidence suggesting that TORC1/Fab1 directly ‘promote’ vacuolar fragmentation. Therefore, we’d like to keep the current statement.
